# Investigation of injury severity in urban expressway crashes: A case study from Beijing

Quan Yuan[1], Xuecai Xu [2]*, Junwei Zhao[3], Qiang Zeng[4]

**1** State Key Laboratory of Automotive Safety and Energy, School of Vehicle and Mobility, Tsinghua University, Beijing, China, **2** School of Civil Engineering and Mechanics, Huazhong University of Science and Technology Wuhan, China, **3** School of Automobile, Chang'an University, Xi'an, China, **4** School of Transportation, South China University of Science and Technology, Guangzhou, China

* xuecai_xu@hust.edu.cn

**Data Availability Statement:** Data cannot be shared publicly because of judicial confidentiality imposed by the Fada Institute of Forensic Medicine & Science and only are restricted within scientific research. For access queries, please contact Mr Wei Ji (Tel: 86-13121198766; Email: joe@cupl.edu.

## Abstract

Urban expressway is the main artery of traffic network, and an in-depth analysis of the crashes is crucial for improving the traffic safety level of expressways. This study intended to address the injury severity of expressways in Beijing by proposing Bayesian ordered logistic regression model. Crash data were collected from urban express rings and expressways in 2015 and 2016. The results showed that crash location, time and crash season are significant variables influencing injury severity. The findings revealed that the proposed model can address the ordinal feature of injury severity, while accommodating the data with small sample sizes that may not adequately represent population characteristics. The conclusions can provide the management departments with valuable suggestions for the injury prevention and safety improvement on the urban expressways.

## Introduction

During the last thirty years, traffic safety has been improved greatly in China, indicating that the improvement of transportation infrastructure and application of advanced transportation technologies have made much progress. However, China is still in top-ranking according to the number of crashes and fatalities. As reported, there were 209,654 injured and 63,772 deaths due to crashes in 2017, and thus there is a long way to go for the traffic safety in China.

Urban expressway is one significant component of traffic network, carrying a large amount of traffic volume and providing convenient service for urban area and long-distance inter-city traffic. Because of heavy traffic and high speed on expressways, the car-following distance is close and lane-changing action is frequent, thus it's more likely to run into rear-end or side crashes, while the crashes may lead to injury or fatality, traffic congestion, and even worse network paralysis if not dealt with immediately. Therefore, the impact of crashes on urban expressways not only causes the severe injury or fatality, but results in network inefficiency of large area, thus it's significant to investigate the influencing factors of crashes on expressways.

During the last decade, there have been a variety of different approaches and perspectives [1–3] presented in safety evaluation, and there are some studies on expressway safety [4–6]. Among them, regression analysis has been widely applied to investigate the relationship

cn) at Fada Institute of Forensic Medicine & Science.

**Funding:** This study was jointly supported by National Natural Science Foundation of China (No: 71801095) to QY and Fundamental Research Fund for the Central Universities [HUST: 2018KFYYXJJ001] to XX.

**Competing interests:** The authors have declared that no competing interests exist.

between injury severity and influencing factors. The widely utilized regression approaches, e.g. linear regression, logistic regression and probit regression, have been accepted by a number of scholars. At early stage, Al-Ghamdi [7] employed binary logistic regression to estimate the influence of accident factors on accident severity. The results found the location and cause of accident were the most significantly associated with severity, and showed that the logistic regression is a promising tool in analyzing safety. Then Yu and Abdel-Aty [8] concluded that binary probit model with Bayesian inference was superior with more significant variables, and the goodness-of-fit improved substantially by considering unobserved heterogeneity in the Bayesian binary probit model. From binary to ordered nature of injury severity levels, one of highly related studies by Park et al. [4] evaluated the influencing factors that contributed to the degree of injury severity sustained in traffic crashes of Korean expressways. Ordered probit, ordered logit and multinomial logit were examined and 16 variables were identified as major contributing factors to the severity of injuries. Michalaki et al. [9] explored the factors affecting motorway accident severity using the generalized ordered logistic regression model in England. The results suggested that the factors positively affecting the severity include the number of vehicles involved, peak-hour traffic time and low visibility. Yoon et al. [10] investigated the influencing factors of injury severity occurred in local bus crashes, and developed a hierarchical ordered model. At the lower level, the influencing factors included vehicle speed, vehicle age, road alignment, surface status, road class and traffic light installation, while at the upper level, pavement, emergent medical environment, traffic rate of compliance, and ratio of elderly in the community were significant. The latest study by Rezapour et al. [11] selected ordered logistic models on crash injury severities of downgrade crashes. The findings provided insights into contributing factors of downgrade crashes in mountainous areas. All the studies have verified that ordered logistic/probit model can be applicable in analyzing the crash injury severity.

Ring road is one important type of urban expressways, and has been widely employed in China. The main function lies in separating the traffic in the downtown area from that in suburban areas, and carrying a large amount of traffic volume to avoid the overloading of urban area. In Beijing, there have been 6 ring roads so far, covering 432 kilometers in total, which constitutes of unique urban structure. As the significant component of urban roadway network, expressways and express rings in Beijing play an important role, and it is necessary to investigate the influencing factors of injury severity to improve the safety level. Therefore, the purpose of this study is to examine the crashes from expressways and express rings in Beijing. The Bayesian ordered logistic model will be proposed to analyze the ordered feature of injury severity by considering crash features, vehicles, roadway conditions and environment comprehensively so that the references can be made to the injury prevention and traffic management for the expressways.

## Data description

The dataset was collected from the real crashes maintained by Beijing Bureau of Traffic Management from 2015 to 2016. The target area in this study was covered by express rings and expressways, including 2nd Ring, 3rd Ring, 4th Ring, 5th Ring and Jing-tong Expressway. There are 166 crashes involved as shown in Fig 1. Since one crash may involve more than one vehicle, some data were double counted. After some invalid data were removed, 133 samples were kept. Four main factors were extracted: the crash features, the vehicle profiles, roadway characteristics and the environment.

According to the data collected from the expressways in Beijing, injury severity is classified into three types, slight (including property damage only), injury (no death) and fatality (1 or more than 1 death). To correspond to the three types, ordered regression model was proposed

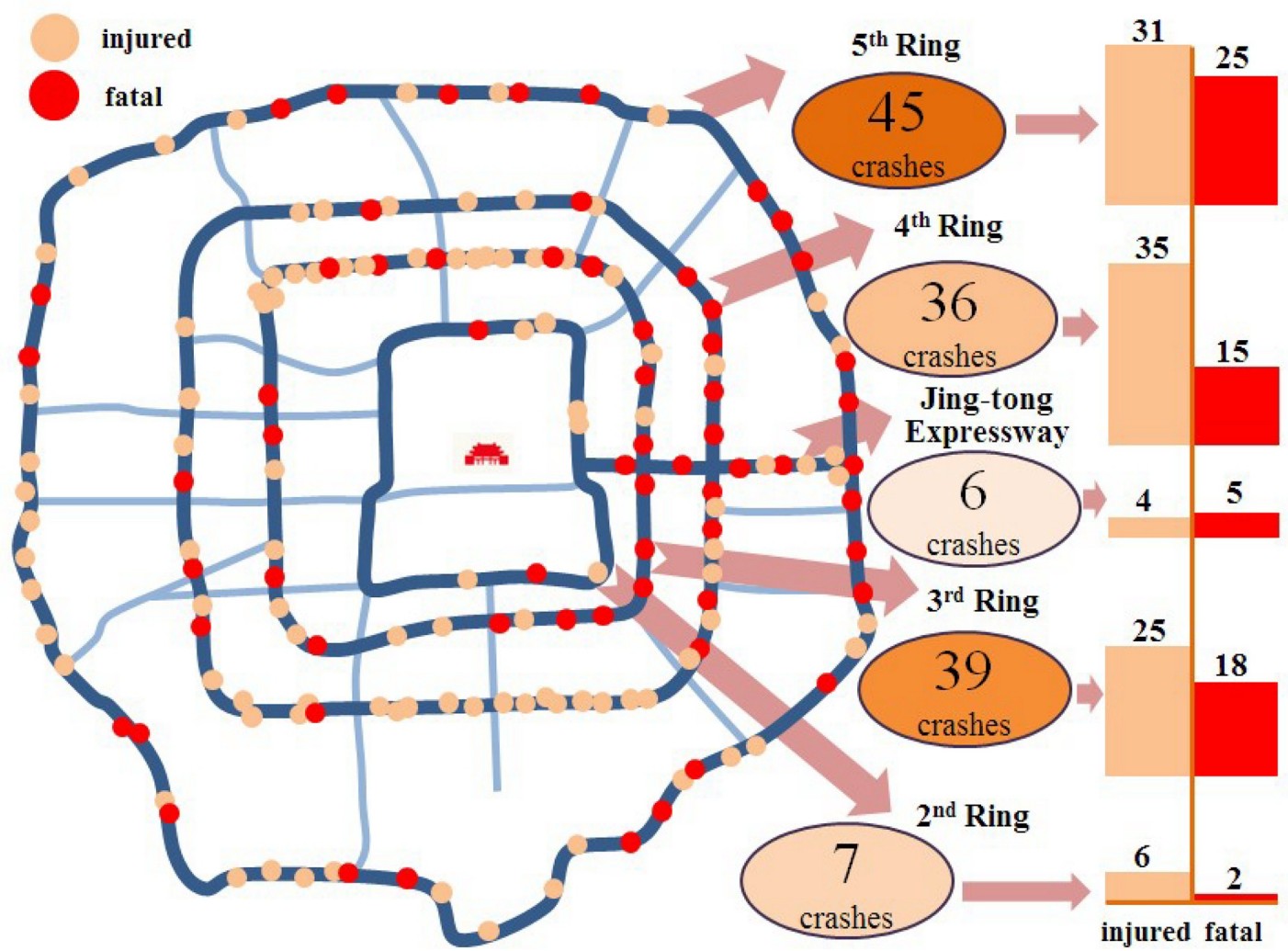

**Fig 1. Study area by selected expressways in Beijing.**

to match with the ordinal feature of injury severity. Therefore, injury severity can be regarded as the dependent variable in the proposed model with slight (1), injury (2) and fatality (3). Moreover, the variables reflecting the crash features, such as crash type, time, date, day, injury location (e.g. segment, ramp or auxiliary lane), etc. are included.

Due to the collection difficulty and privacy, the drivers' personal status, e.g. age, gender, action, and conditions, were not provided, thus the dataset in this study mainly concentrates on the non-behavioral variables.

According to the vehicles involved during the injury, the explanatory variables reflecting the vehicle profiles include vehicle type and vehicle action. Furthermore, the crash data collected involve either two vehicles or more than two vehicles, in which the vehicle with main responsibility is named as vehicle 1, and those with minor responsibility is as vehicle 2. According to the data collected, crashes with two vehicles account for over 90%, thus the classification is reasonable.

Since express rings and expressways are the objects, the roadway characteristics contain the number of ring roads, and roadway surface (e.g. dry, wet (rain/snow), and others), while the crash environment extracts the weather and season.

In order to evaluate the proposed models in STATA software, the categorical variables are digitalized, and all the variables collected are listed and summarized in Table 1 with dependent and categorical variables before, and the descriptive statistics of the indicator variables in the following.

## Methodology

Generally the standard ordered regression logistic model employs unobservable variable z to represent the latent variable, which can be considered as the foundation of modeling the ordinal feature of the data, thus the discrete injury severity levels can be assumed to be concerned with the continuous latent variable. The specification of the latent variable for each observation can be expressed as [12]:

$$z = \beta X_i + \varepsilon_i \tag{1}$$

where $\beta$ represents the vector of estimated coefficients, $X_i$ denotes the vector of influencing variables for each crash observation, and $\varepsilon_i$ is the random error term. With the Eq (1), the observed ordinal injury severity levels (y) can be described as follows:

$$\begin{cases} y = 1 \text{ if } z \leq \mu_0 \text{(Slight)} \\ y = 2 \text{ if } \mu_0 < z \leq \mu_1 \text{(Injury)} \\ y = 3 \text{ if } \mu_1 < z \leq \mu_2 \text{(Fatality)} \end{cases} \tag{2}$$

where $\mu_i$ is the threshold that defines the injury severity y. Given the value of $X_i$, the probability that the injury severity of individual i belongs to each category is the followings:

$$\begin{cases} P(y = 1) = \emptyset(-\beta X) \\ P(y = 2) = \emptyset(\mu_1 - \beta X) - \emptyset(-\beta X) \\ P(y = 3) = \emptyset(\mu_2 - \beta X) - \emptyset(\mu_1 - \beta X) \end{cases} \tag{3}$$

where $\emptyset(\cdot)$ is the standard logistic cumulative distribution function. The parameter estimation can be realized using the log-likelihood approach, and the likelihood function for the ordered logit model can be expressed as:

$$LL = \sum_{n=1}^{N} \sum_{i=1}^{I} \delta_{in} LN[\emptyset(\mu_1 - \beta X_n) - \emptyset(\mu_{1+1} - \beta X_n)] \tag{4}$$

where $\delta_{in}$ is equal to 1 if the observed discrete outcome is i, and zero otherwise. The odds of the crash outcome i can be described as:

$$\frac{P(y = i)}{1 - P(y = i)} = \exp(\beta_0 + \beta_1 X) = e^{\beta_0}(e^{\beta_1})^X \tag{5}$$

However, the injury severity levels may vary across spatial location, e.g. severity levels may be higher at some expressways while lower at others. In such cases, the extent of the effect of severity levels may be different. At this point, in statistical terms, there exists within-individual homogeneity and between-individual heterogeneity in the hierarchically structured data, and multilevel modeling approach provides an appropriate analytical framework to deal with the spatial issue. In this study, the basic cross-sectional ordered logistic model as the first level, and then the model development expands the basic model by adding the panel data to explain the between-expressway heterogeneity, which specifies the random intercept sigma$^2$at the expressway level.

Due to this, in this study Bayesian estimation approach is employed for the multilevel logistic model. For Bayesian inference, the likelihood function is used to update the prior

**Table 1. Summary of the parameters.**

| Variable | Description | Count (proportion) | | | |
|---|---|---|---|---|---|
| **i) Dependent variables** | | | | | |
| **Injury severity** | 1-slight | 8(6.0%) | | | |
| | 2-injury | 63(47.4%) | | | |
| | 3-fatality | 62(46.6%) | | | |
| **ii) Categorical variables** | | | | | |
| **Crash type** | 1-Rear-end | 47(35.3%) | | | |
| | 2-Single vehicle | 18(13.5%) | | | |
| | 3-Sidewipe | 17(12.8%) | | | |
| | 4-Head-on | 8(6.0%) | | | |
| | 5-Others | 43(32.4%) | | | |
| **Crash location** | 1-Segment | 79 (59.4%) | | | |
| | 2-On/off ramp | 11(8.3%) | | | |
| | 3-Auxiliary lane | 43(32.3) | | | |
| **Crash season** | 1-Spring | 14(10.5%) | | | |
| | 2-Summer | 35(26.3%) | | | |
| | 3-Autumn | 56(42.1%) | | | |
| | 4-Winter | 28(21.1%) | | | |
| **Vehicle 1 type** | 1-Motor/ebike | 22(16.5%) | | | |
| | 2-Car | 34(25.6%) | | | |
| | 3-Pickup/van | 36(27.0%) | | | |
| | 4-Heavy truck | 25(18.8%) | | | |
| | 5-Unknown | 16(12.1%) | | | |
| **Vehicle 1 action** | 1-Striking | 68(51.1%) | | | |
| | 2-Struck | 46(34.6%) | | | |
| | 3-Others | 19(14.3%) | | | |
| **Vehicle 2 type** | 1-Motor/ebike | 25(18.8%) | | | |
| | 2-Car | 25(18.8%) | | | |
| | 3-Pickup/van | 28(21.0%) | | | |
| | 4-Heavy truck | 18(13.5%) | | | |
| | 5-Unknown | 37(27.9%) | | | |
| **Vehicle 2 action** | 1-Striking | 62(46.6%) | | | |
| | 2-Struck | 25(18.8%) | | | |
| | 3-Others | 46(34.6%) | | | |
| **Road surface** | 1-Dry | 94(70.7%) | | | |
| | 2-Wet (rain/snow) | 14(10.5%) | | | |
| | 3-Others | 25(18.8%) | | | |
| **Weather condition** | 1-Clear | 89(66.9%) | | | |
| | 2-Cloudy | 9(6.8%) | | | |
| | 3-Rain/snow | 9(6.8%) | | | |
| | 4-Other | 26(19.5%) | | | |
| | | **Mean** | **S.D.** | **Min.** | **Max.** |
| **iii) Indicator variables** | | | | | |
| Time | Daytime (0) or nighttime (1) | 0.54 | 0.50 | 0 | 1 |
| Period | Offpeak (0) or peak (1) | 0.14 | 0.35 | 0 | 1 |
| Week | Weekday (0) or weekend (1) | 0.30 | 0.46 | 0 | 1 |

distributions and achieve the posterior distribution of parameters. Assume θ to denote the parameters to be estimated, the posterior distribution of θ can be computed as:

$$\pi(\theta|y) = \frac{f(y|\theta)\pi(\theta)}{\int_\theta (y|\theta)\pi(\theta)d\theta} \propto f(y|\theta)\pi(\theta) \tag{6}$$

where y = {$y_1$,. . .,$y_i$,. . .$y_n$} represents the observed outcomes, $\pi(\theta)$ denotes the prior distribution of θ, $f(y|\theta)$ denotes the sampling distribution, $\int_\theta (y|\theta)\pi(\theta)d\theta$ represents the marginal distribution of y, and $\pi(\theta|y)$ denotes the posterior distribution of θ. It can be seen that the Bayesian inference provides a flexible framework to integrate the prior knowledge of the data with the parameter estimation process. This is especially important for data with small sample sizes that may not adequately represent population characteristics [13]. More details about the ordered logistic model and Bayesian inference can be referred to [4, 11–13].

For model comparison, as provided by many other studies with the Bayesian inference [14, 15, 16], the Deviance Information Criterion (DIC) is used to evaluate the proposed Bayesian ordered logistic regression model, whereas Akaike Information Criterion (AIC) and Bayesian Information Criterion (BIC) are employed to evaluate the goodness-of-fit about ordered logistic regression model, thus, multilevel ordered logistic model is employed by considering time as the 2nd level within Bayesian framework so as to make the comparison equally. Therefore, DIC is used to compare the models abovementioned:

$$DIC = D(\bar{\theta}) + 2p_D = \bar{D} + p_D \tag{7}$$

where $D(\bar{\theta})$ is the deviance evaluated at $\bar{\theta}$, the posterior mean of the parameter of interest, $p_D$ is the effective number of parameter in the model, and $\bar{D}$ is the posterior mean of the deviance statistic $D(\bar{\theta})$. The lower the DIC, the better the model fits. Generally speaking, differences in DIC of more than 10 definitely rule out the model with the higher DIC; differences between 5 and 10 are considered substantial, while the difference less than 5 indicates that the models are not statistically different from each other.

## Results and discussion

Based on all the variables selected from the 133 crash cases, the characteristics of the crashes and correlation among main factors can be examined. In this study, STATA software was employed to store and analyze the data. The correlation test showed that there is high correlation between road surface and weather condition, vehicle 2, vehicle 2 action and vehicle 1 action. Thus, in the final results the variables may not occur at the same time.

The Bayesian multilevel ordered logistic and Bayesian ordered logistic regression model were developed to examine the injury severity in urban expressways. For Bayesian inference, the first 2,500 iterations in each distribution were discarded as burn-in, and then 10,000 iterations were conducted for each distribution of 12,500 for each parameter. The models convergence was monitored by the ratios of Monte Carlo errors relative to the respective standard deviation of the estimates, which should be less than 0.05. The final model is presented in Table 2.

Shown from Table 2, for both models, crash location, time and crash season are significant variables influencing injury severity. The log marginal likelihood of Bayesian ordered logistic model (-124.731) is close to that of multilevel ordered logistic model (-123.916), while the difference of DIC values are less than 5, indicating that the goodness-of-fit of Bayesian inference is not significantly different from each other, but DIC value of proposed model is smaller, thus the following explanation would concentrate on the Bayesian ordered logistic regression model.

In Table 2, there are three significant variables influencing injury severity in urban expressways. Crash location is negatively associated with injury severity, implying that compared to

**Table 2. Parameter estimates for the proposed models.**

| Variable | Bayesian multilevel ordered logistic | | | | Bayesian ordered logistic | | | |
|---|---|---|---|---|---|---|---|---|
| | Mean | Std. Dev. | MCSE | 95% BCI | Mean | Std. Dev. | MCSE | 95% BCI |
| Crash location | -0.467* | 0,201 | 0.013 | (-0.863,-0.067) | -0.409* | 0.207 | 0.009 | (-0.831,-0.021) |
| Time | 0.896* | 0.365 | 0.026 | (0.205, 1.637) | 1.000* | 0.372 | 0.025 | (0.224,1.724) |
| Crash season | 0.501* | 0.211 | 0.025 | (0.102,0.910) | 0.554* | 0.209 | 0.009 | (0.160,0.964) |
| Cut1 | -2.127 | 0.798 | | | -1.785 | 0.773 | | |
| Cut2 | 1.184 | 0.765 | | | 1.494 | 0.732 | | |
| Sigma$^2$ | 0.277 | 0.460 | | | | | | |
| **Goodness-of-fit** | | | | | | | | |
| **No. of observations** | 133 | | | | 133 | | | |
| **DIC** | 223.312 | | | | 222.297 | | | |
| **Log marginal likelihood** | -123.916 | | | | -124.731 | | | |

Note: Std. Dev. = Standard Deviation; MCSE = Monte Carlo Standard Error; BCI = Bayesian credible interval;

* denotes significance at 95% confidence interval.

injury at segment, the severity is slighter at ramp and auxiliary lanes. The reason is such that although at ramp and auxiliary lanes more lane changing and more conflicts occur, the speed at segment is much higher, thus leading to more severe injury. Various studies [8, 17] have verified that excessive speeding is crucial for the injury severity on freeway segments.

The second significant variable time is positively concerned with injury severity, indicating that injury at nighttime is more severe than that in the daytime. Ususally at nighttime the traffic volume on expressways is lower than that in the daytime, but the speed is much higher, so the probability of running into severe injury is higher, which is in line with Jang et al. [18] and Yuan and Chen [19].

Another significant variable crash season is positively related to injury severity, meaning that the probability of injury in winter is higher than in the rest seasons. This is uniform with the basic knowledge, since the weather in Beijing belongs to temperate monsoon climate by featuring short spring and autumn, hot summer and cold winter. In winter when there is heavy snow, the probability of severe injury is increased to a large extent. Although the injury accounts for a high proportion in autumn in Table 2, the severity in winter is still the worst, sometimes causing a series of crashes and fatalities on expressways, which has been examined by some studies [20–22].

According to the results obtained, from an empirical point of view, for the department of traffic management, speed limit sign should be clearly established at certain distance on expressway segment, and electronic velocity measurement combined with dynamic message sign (DMS) should be made at long segment so that excessive speeding would be reduced to lessen the injury severity; At nighttime the lighting facilities or devices should be kept under good conditions to help the expressway users increase the sight and more alert facilities, such as voice warning, flashing lights, etc., should be set up to avoid the driving fatigue at night; the winter season increases the injury severity, thus one way of increasing the safety is to remove the ice/snow with facilities as soon as possible, and guarantee the roadway conditions clearly.

## Conclusions

A variety of studies have concerned the injury severity at different locations, but not many have been explored with respect to the urban expressways. In this paper we proposed ordered logistic regression model within Bayesian framework to address the injury severity of

expressways in Beijing. This method permits to address the ordinal feature of injury severity, and the inference is highlighted in a straightforward manner from the Bayesian point of view. Moreover, the Bayesian inference allows for an easy derivation of the posterior credible intervals, which provides a clear measure for data with small sample sizes that may not adequately represent population characteristics. The suitability of the method is illustrated with the dataset in Beijing from 2015 to 2016.

This study adds to the injury severity in three aspects. First, the Bayesian ordered logistic regression model in the injury severity analysis can accommodate the data with small sample sizes that may not adequately represent population characteristics; Second, the goodness-of-fit of the proposed model performs no difference from corresponding multilevel ordered logistic model, while addressing the odernal feature of injury severity precisely; Finally, the results can provide some potential insights in expressway safety improvement.

One concern is that the data collected may be the drawback, and if more comprehensive data (e.g. drivers' status, motorcyclists, 3 to 5 years), the preciseness of injury severity may be better reflected. Another issue is that travel speed may be significantly associated with traffic safety [23], and although speed limits have been collected in this study, they are not reflected from the actual modeling process. Although time was considerd as the 2$^{nd}$ level in Bayesian multilevel ordered logistic model, the two-year's data may not address the time-series feature of injury severity. Therefore, an extension of the present injury severity problem could be dealt with by time-series data more than three years combing with cross-sectional data within Bayesian framework, in this way the spatial-temporal issue can be addressed [24], which is our next-step work. This will broaden the scope of injury severity in expressways, and can provide a much safer expressway environment.

## Author Contributions

**Conceptualization:** Quan Yuan, Qiang Zeng.

**Data curation:** Junwei Zhao.

**Formal analysis:** Xuecai Xu.

**Funding acquisition:** Quan Yuan.

**Methodology:** Xuecai Xu.

**Validation:** Xuecai Xu.

**Writing – original draft:** Quan Yuan.

**Writing – review & editing:** Xuecai Xu.

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
