## [Decision Letter · Decision Letter 0]

5 Nov 2019

PONE-D-19-29212

Investigation of Injury Severity in Urban Expressway Crashes: A Case Study from Beijing

PLOS ONE

Dear Dr. Xu,

Thank you for submitting your manuscript to PLOS ONE. After careful consideration, we feel that it has merit but does not fully meet PLOS ONE’s publication criteria as it currently stands. Therefore, we invite you to submit a revised version of the manuscript that addresses the points raised during the review process.

We would appreciate receiving your revised manuscript by Dec 20 2019 11:59PM. To enhance the reproducibility of your results, we recommend that if applicable you deposit your laboratory protocols in protocols.io, where a protocol can be assigned its own identifier (DOI) such that it can be cited independently in the future. For instructions see: http://journals.plos.org/plosone/s/submission-guidelines#loc-laboratory-protocols

We look forward to receiving your revised manuscript.

Kind regards,

Feng Chen

Academic Editor

PLOS ONE

Journal Requirements:

2.

We suggest you thoroughly copyedit your manuscript for language usage, spelling, and grammar. If you do not know anyone who can help you do this, you may wish to consider employing a professional scientific editing service.  

https://doi.org/10.1061/(ASCE)0733-947X(2009)135:1(18)

https://doi.org/10.1155/2019/8521649

In your revision ensure you cite all your sources (including your own works), and quote or rephrase any duplicated text outside the methods section. Further consideration is dependent on these concerns being addressed.

Reviewers' comments:

Reviewer's Responses to Questions

**Comments to the Author**

1. Is the manuscript technically sound, and do the data support the conclusions?

Reviewer #1: Yes

Reviewer #2: Yes

2. Has the statistical analysis been performed appropriately and rigorously? 

Reviewer #1: Yes

Reviewer #2: Yes

3. Have the authors made all data underlying the findings in their manuscript fully available?

Reviewer #1: Yes

Reviewer #2: Yes

4. Is the manuscript presented in an intelligible fashion and written in standard English?

Reviewer #1: Yes

Reviewer #2: Yes

5. Review Comments to the Author

Reviewer #1: This is an interesting study that focused on the crash injury severity. The overall analytical works and results are sounding. The followings are my comments and suggestions.

1. Did the authors also obtaining the Property Damage Only crashes? It seems that only injury and above severity crashes were collected. At least the readers should be informed with this kind of information regarding the quantity and the proportion of the severe crashes.

2. For the variables considered, as a recent study claims that speed has great influence on the urban expressway crashes. (Rongjie Yu*, Mohammed Quddus, Xuesong Wang, Kui Yang, 2018. Impact of data aggregation approaches on the relationships between operating speed and traffic safety. Accident Analysis & Prevention 120, 304-310. ) However, from Table 1 it seems the authors did not consider this type of parameter. Please justify.

3. Bayesian inference different from the traditional statistical model that they do not have a likelihood function. However, the authors listed both likelihood values for the two types of models. Please justify this issue.

Reviewer #2: In the present study, a Bayesian ordered logistic regression model is proposed to investigate the impact factors for injury severity in urban expressway crashes. The content is well organized. However, I have several concerns as follows.

1. The manuscript employed the ordered logistic model. However, in P2 ln 32, according to the literature review, the authors concluded that “ordered logistic/probit model” is applicable. I wonder why the ordered logistic model is selected?

2. Please double check the crash counts in Figure 1. The authors claimed that 133 samples were kept, However, there were 134 crashes in the figure (46+36+6+39+7=134). By the way, it would be better to give a brief description about the reason why 33 records were deleted.

3. P3 ln 17, it is easy to see the difference between injury crash and fatality crash. But how about the injury severity type of slight. And I wonder is non-injury crashes included in this study. If not, please give an explanation.

4. P5 table 1, the “S.D.” is not provided. And in order to provide a comprehensive data description, I would recommend the authors provide an injury-specified data description.

5. P6 ln 27-30, please double check the log marginal likelihood value. In general, higher log likelihood, or say lower absolute value, indicates better goodness-of-fit. And it is really confusing that how can the authors get the log marginal likelihood value for a model estimated using maximum likelihood estimation approach. In addition, there seems no significant difference between the estimated coefficients in the two models.

6. It seems that only three variables entered the final model. It would be great if the authors could conduct a comparison study among the proposed ordered logistic model, and some other state-of-art models, such as multinomial logit model, mixed logit model, etc.

7. There are a lot of typos and grammar errors. I would highly recommend the authors to get editing help from someone with full professional proficiency in English. For example, P1 ln45, “fatality” shall be “fatalities”; P1, ln 46, P2, ln 2 “thus” is not a conjunction. Please use “and thus”; P2, ln 13 “tha”; P5 ln 13, “thresholds” shall be “threshold”; P6 ln33, “inury”.

6. PLOS authors have the option to publish the peer review history of their article (what does this mean?). If published, this will include your full peer review and any attached files.

Reviewer #1: No

Reviewer #2: No

---

## [Author Response · Author response to Decision Letter 0]

18 Nov 2019

Please refer to the Response Letter.

---

## [Decision Letter · Decision Letter 1]

23 Dec 2019

PONE-D-19-29212R1

Investigation of Injury Severity in Urban Expressway Crashes: A Case Study from Beijing

PLOS ONE

Dear Dr. Xu,

Thank you for submitting your manuscript to PLOS ONE. After careful consideration, we feel that it has merit but does not fully meet PLOS ONE’s publication criteria as it currently stands. Therefore, we invite you to submit a revised version of the manuscript that addresses the points raised during the review process.

Whilst the performance has been compared to other existing methods, please also check whether the methods chosen for comparison are state-of-the-art. 

We would appreciate receiving your revised manuscript by Feb 06 2020 11:59PM. To enhance the reproducibility of your results, we recommend that if applicable you deposit your laboratory protocols in protocols.io, where a protocol can be assigned its own identifier (DOI) such that it can be cited independently in the future. For instructions see: http://journals.plos.org/plosone/s/submission-guidelines#loc-laboratory-protocols

We look forward to receiving your revised manuscript.

Kind regards,

Feng Chen

Academic Editor

PLOS ONE

Reviewers' comments:

Reviewer's Responses to Questions

**Comments to the Author**

1. If the authors have adequately addressed your comments raised in a previous round of review and you feel that this manuscript is now acceptable for publication, you may indicate that here to bypass the “Comments to the Author” section, enter your conflict of interest statement in the “Confidential to Editor” section, and submit your "Accept" recommendation.

Reviewer #1: All comments have been addressed

Reviewer #2: (No Response)

2. Is the manuscript technically sound, and do the data support the conclusions?

Reviewer #1: Yes

Reviewer #2: Yes

3. Has the statistical analysis been performed appropriately and rigorously? 

Reviewer #1: Yes

Reviewer #2: Yes

4. Have the authors made all data underlying the findings in their manuscript fully available?

Reviewer #1: Yes

Reviewer #2: Yes

5. Is the manuscript presented in an intelligible fashion and written in standard English?

Reviewer #1: Yes

Reviewer #2: Yes

6. Review Comments to the Author

Reviewer #1: My previous concerns have been well addressed. And I have no further comments or suggestions. The paper can be published as its current format.

Reviewer #2: Thanks for the authors' carefully addressing my comments in last round-review. However, there do have several problems in the revised manuscript, as follows.

(1) Page 5, Eq (2), the constraint on z value shall be a combination of one open interal and one closed interval;

(2) Page 5, ln16, the application of multilevel ordered logistic model shall be better motivated and described. In the result analysis, the multilevel model performs almost the same as the ordered model.

(3) In Table 2, where is the Coef and Std Err as described in the note information.

7. PLOS authors have the option to publish the peer review history of their article (what does this mean?). If published, this will include your full peer review and any attached files.

Reviewer #1: No

Reviewer #2: No

---

## [Author Response · Author response to Decision Letter 1]

24 Dec 2019

Please refer to Resonses to reviewers' comments.

---

## [Decision Letter · Decision Letter 2]

2 Jan 2020

Investigation of Injury Severity in Urban Expressway Crashes: A Case Study from Beijing

PONE-D-19-29212R2

Dear Dr. Xu,

We are pleased to inform you that your manuscript has been judged scientifically suitable for publication and will be formally accepted for publication once it complies with all outstanding technical requirements.

With kind regards,

Feng Chen

Academic Editor

PLOS ONE

Additional Editor Comments (optional):

Reviewers' comments:

Reviewer's Responses to Questions

**Comments to the Author**

1. If the authors have adequately addressed your comments raised in a previous round of review and you feel that this manuscript is now acceptable for publication, you may indicate that here to bypass the “Comments to the Author” section, enter your conflict of interest statement in the “Confidential to Editor” section, and submit your "Accept" recommendation.

Reviewer #2: All comments have been addressed

2. Is the manuscript technically sound, and do the data support the conclusions?

Reviewer #2: Yes

3. Has the statistical analysis been performed appropriately and rigorously? 

Reviewer #2: Yes

4. Have the authors made all data underlying the findings in their manuscript fully available?

Reviewer #2: Yes

5. Is the manuscript presented in an intelligible fashion and written in standard English?

Reviewer #2: Yes

6. Review Comments to the Author

Reviewer #2: Thanks for the authors' effort. My previous concerns have been addressed. And I have no further comments or suggestions.

7. PLOS authors have the option to publish the peer review history of their article (what does this mean?). If published, this will include your full peer review and any attached files.

Reviewer #2: No

---

## [Editor Report · Acceptance letter]

6 Jan 2020

PONE-D-19-29212R2 

Investigation of Injury Severity in Urban Expressway Crashes: A Case Study from Beijing 

Dear Dr. Xu:

I am pleased to inform you that your manuscript has been deemed suitable for publication in PLOS ONE. Congratulations! Your manuscript is now with our production department. 

With kind regards,

on behalf of

Dr. Feng Chen 

Academic Editor

PLOS ONE